# Bluetooth Low Energy Mesh Networks: Survey of Communication and Security Protocols

**DOI:** 10.3390/s20123590

**Published:** 2020-06-25

**Authors:** Muhammad Rizwan Ghori, Tat-Chee Wan, Gian Chand Sodhy

**Affiliations:** 1School of Computer Sciences, Universiti Sains Malaysia, Penang 11800, Malaysia; mrizwanghori@student.usm.my (M.R.G.); sodhy@usm.my (G.C.S.); 2National Advanced IPv6 Centre, Universiti Sains Malaysia, Penang 11800, Malaysia

**Keywords:** bluetooth low energy, BLE, wireless mesh networks, IoT security

## Abstract

Bluetooth Low Energy (BLE) Mesh Networks enable flexible and reliable communications for low-power Internet of Things (IoT) devices. Most BLE-based mesh protocols are implemented as overlays on top of the standard Bluetooth star topologies while using piconets and scatternets. Nonetheless, mesh topology support has increased the vulnerability of BLE to security threats, since a larger number of devices can participate in a BLE Mesh network. To address these concerns, BLE version 5 enhanced existing BLE security features to deal with various authenticity, integrity, and confidentiality issues. However, there is still a lack of detailed studies related to these new security features. This survey examines the most recent BLE-based mesh network protocols and related security issues. In the first part, the latest BLE-based mesh communication protocols are discussed. The analysis shows that the implementation of BLE pure mesh protocols remains an open research issue. Moreover, there is a lack of auto-configuration mechanisms in order to support bootstrapping of BLE pure mesh networks. In the second part, recent BLE-related security issues and vulnerabilities are highlighted. Strong Intrusion Detection Systems (IDS) are essential for detecting security breaches in order to protect against zero-day exploits. Nonetheless, viable IDS solutions for BLE Mesh networks remain a nascent research area. Consequently, a comparative survey of IDS approaches for related low-power wireless protocols was used to map out potential approaches for enhancing IDS solutions for BLE Mesh networks.

## 1. Introduction

Bluetooth Low Energy (BLE) is an increasingly prevalent Wireless Ad-Hoc Network (WAHN) technology for battery-powered Internet of Things (IoT) devices [1]. The BLE standard was introduced by the Bluetooth Special Interest Group (SIG) in Bluetooth version 4.0, and subsequently enhanced in versions 4.2 and 5. Initially, BLE 4.x adopted the legacy Bluetooth Personal Area Network (PAN) model for multi-hop communications and the interconnection of networks. BLE 5 intends to address these inadequacies via the implementation of pure mesh topology to provide enhanced network coverage, inter-network connectivity, and improved security [2]. In this paper, we surveyed the most recent BLE-based communication protocols and related security issues in order to understand the current state of BLE Mesh protocol development and open research areas. This research has mostly addressed new protocols and issues discovered since the publication of the existing surveys. Additionally, we have analyzed both BLE based communication protocols and security-related concerns to address mesh network issues and BLE 5 specific security weaknesses, respectively.

The vast majority of BLE-based applications still assume a star network topology while using BLE Beacons in broadcast mode [3,4,5,6,7,8,9]. To enhance the coverage of BLE 4 networks, hybrid mesh topologies extend the master-slave piconet concept into various interconnected scatternets via the fusion of star and mesh links [10]. Nonetheless, reliability and scalability remain an issue. In contrast, a pure mesh topology removes the master–slave limitation by making nodes peer with each other to form scalable networks. Nonetheless, there is a lack of research on the implementation of BLE 5 pure mesh topologies to date. In addition, the proposed protocols lack multicasting and topology auto-configuration capabilities [11].

The BLE specifications has adopted various useful security features. However, recent security exploits have highlighted the vulnerabilities in existing BLE security features [12]. Consequently, security features must be supplemented with strong Intrusion Detection Systems (IDS) to detect zero-day attacks. Because the BLE 5 Mesh protocol expands the reach of the network significantly, the potential for intrusion also increases proportionally. Moreover, it is necessary to reference existing IDS approaches adopted by other low-power wireless network technologies for comparison due to the lack of research into suitable IDS for BLE Mesh networks. The issues and features of the competing solutions will be analyzed to determine relevant solutions that can be adapted for BLE Mesh networks.

The organization of the rest of the paper is as follows: Section 2 is a brief overview of the BLE Mesh System Architecture, while Section 3 surveys various BLE Mesh Communications Protocols. Section 4 discusses various BLE Mesh Security Issues and proposed solutions, while Section 5 explores IDS for related WSN technologies, in order to determine relevant approaches to facilitate the development of IDS for BLE Mesh networks. Finally Section 6 summarizes the discussions for the paper.

## 2. BLE Mesh System Architecture

The BLE Mesh System Architecure is an overlay on top of the BLE Network Stack. A brief overview of the core BLE Network Stack and BLE Mesh Layers is provided to define various terms and concepts used in the rest of the paper.

### 2.1. BLE Network Stack

The BLE Network Stack comprises of three primary layers, namely, Host, Controller and Physical/Radio Layers [13]. A brief explanation is provided in this section as a reference for subsequent sections.

The Host layer is positioned just below the application layer, and it is embedded with several non-real-time network and transport protocols for communications between applications on different devices. The modules in this layer include the Generic Access Profile (GAP), Generic Attribute Profile (GATT), Security Manager (SM), Attribute Protocol (ATT), and Logical Link Control and Adaptation Protocol (L2CAP).

The Controller layer implements the BLE Link Layer (LL) protocols (low-level and real-time). Apart from handling control procedures and physical layer interfacing through the Host Control Interface (HCI), BLE LL performs packet reception, schedules transmissions, and ensures data delivery [14].

The physical layer is the lowest layer responsible for the transmission of wireless signals. BLE operates on the 2.4 GHz ISM (Industrial, Scientific, and Medical) frequency band with 40 narrowband channels (2 MHz bandwidth) split into 3 (Ch. 37–39) Advertising Channels (AC) and 37 (Ch. 0–36) Data Channels (DC) [15]. The ACs are used for device discovery, connection establishment, and broadcast messages transmission. In contrast, DCs enable two-way data transfer among connected devices and use Adaptive Frequency Hopping (AFH) for subsequent communications.

### 2.2. BLE Communication Profiles

A brief overview of BLE Communication Profiles is provided in this section.

The Generic Access Profile (GAP) defines a general topology for how BLE devices interconnect [16]. A BLE device might act as a Broadcaster (advertises but does not allow connections), an Observer (sees advertisements but does not initiate a connection), a Peripheral (advertises and accepts connections), or a Central (sees advertisements and initiate connections) [2]. BLE implements connectionless communications via the use of advertisements (termed beacons) between Broadcasters and Observers, as well as connection-oriented communications between Peripheral and Central devices.

For example, a BLE Peripheral device initially advertises its presence via broadcasts, while the receiving device, a mobile phone acting as the Central, establishes a connection with the Peripheral for two-way communications. After connection establishment, the Central will act as the master, while the Peripheral will act as the slave. Devices may also implement multiple roles to support more complex topologies.

The Generic Attribute (GATT) profile defines how data transfer takes place after the establishment of a dedicated connection by the GAP. In addition, it specifies roles for the nodes, where one acts as the client and the other as the server.

The Security Manager defines the methods for device pairing and key distributions. It offers services to other layers for secure connections and data transfer between the devices [17].

The Attribute protocol defines the role of clients and servers. A client sends requests for reading and writing available attributes (data) that are stored in the server, while the server is responsible for storing the attributes and making them available to the client [17].

The L2CAP profile provides connection-oriented and connectionless data services to the upper layer, along with multiplexing, segmentation, and reassembly capabilities [18].

### 2.3. BLE Mesh Layers

The BLE Mesh System Architecture is defined on top of the BLE core specifications [17], as shown in Figure 1. In the figure, the Bearer Layer leverages the BLE Network Stack Host protocols to support its operation.

The Model layer consists of models that define operations based on usage scenarios. These models are defined either as Bluetooth Mesh Model Specification (Bluetooth SIG defined models are discussed in the Appendix A) or by vendors (vendor models). The models are identified via 16-bit and 32-bit unique identifiers defined by the Bluetooth SIG and vendor, respectively.

The Foundation Model layer defines the states, messages, and models that are required to configure and manage mesh networks for particular scenarios. There are two sets of models described in the Bluetooth SIG Specification, i.e., the Configuration Client and Server model, and the Health Client and Server model.

The Access Layer defines how upper layers can utilize the Upper Transport Layer. Moreover, it is responsible for specifying the application data format and implementing encryption/decryption functions. Finally, it verifies that incoming data contain the correct network and application keys before forwarding the data to the upper layers.

The Upper Transport layer is responsible for performing encryption, decryption, and authentication of application data and it offers access messages confidentiality. Moreover, it defines the control messages for coordinating the transport layer functions between nodes.

The Lower Transport layer defines the segmentation and reassembly of upper layer messages into various lower-layer protocol data units. In addition, it is responsible for the management of segmentation and reassembly control messages.

The Network layer handles the addressing, formatting, encryption, and authentication for data transmissions. Message forwarding and dropping decisions are also the responsibility of this layer.

The Bearer Layer defines the transmission mechanism for messages. Currently, there are two bearers available in the latest BLE 5 Mesh specifications, i.e., Advertising bearer and GATT bearer.

## 3. BLE Mesh Communication Protocols

We have reviewed the most recent BLE-based mesh communication protocols, as shown in Figure 2. A critical discussion of the cited works, along with the pros and cons and open issues, is presented after the overview of the various protocols in each category. This discussion is summarized in Table 1.

### 3.1. Message Forwarding Paradigms

A brief description of three common message forwarding paradigms adopted by various multi-hop forwarding protocols is provided here to help the reader to better understand the taxonomy in Figure 2.

#### 3.1.1. Reactive (On-Demand) Protocols

Reactive forwarding protocols obtain information regarding destination nodes via received messages [19]. Each forwarding table entry only lasts for a certain amount of time. If no traffic for a particular destination were encountered within the specified period, the entry will be discarded. A new route discovery process will be initiated if requested by the sending node [20,21]. Examples of reactive forwarding protocols are Ad-hoc On-Demand Distance Vector (AODV) [22] and Dynamic Source Control Routing (DSR) [23].

#### 3.1.2. Proactive (Table Driven) Protocols

Proactive forwarding protocols maintain explicit forwarding table entries for all nodes, whether they are active destinations or not. The Bellman–Ford Algorithm is commonly used to maintain feasible paths to the respective nodes, and data can be forwarded immediately to a destination without delay. Examples of proactive forwarding protocols include Babel [24], Optimized Link State Routing (OLSR) [25], Destination Sequenced Distance Vector (DSDV) [26], Distance Routing Effect Algorithm for Mobility (DREAM) [27], and Better Approach To Mobile Adhoc Networking (BATMAN) [28].

#### 3.1.3. Cluster-Based Protocols

Scatternet is a type of cluster-based forwarding protocol introduced in BLE 4.1 to support multihop communications. Cluster-based forwarding protocols were originally developed for mobile ad-hoc networks. It divides the nodes of a network into several overlapping disjoint clusters [29]. Each cluster has an elected cluster head that is responsible for the maintenance of cluster memberships, which is then used for inter-cluster path discovery. The grouping of nodes into clusters reduces flooding during the path discovery process. Moreover, the protocol keeps track of any unidirectional links for inter-cluster and intra-cluster forwarding. Examples of such protocols include Two-Tier Data Dissemination Protocol (TTDD) [30], Ring Routing [31], Energy Efficient Secured Ring Routing (E2SR2) [32], Intelligent and Secured Fuzzy Clustering Algorithm Using Balanced Load Sub-Cluster Formation (ISFC-BLS) [33], Scalable Energy Efficient Clustering Hierarchy protocol (SEECH) [34], and Multi-Objective Fuzzy Clustering Algorithm (MOFCA) [35].

### 3.2. BLE Hybrid Mesh Protocols

Traditional Bluetooth networks were based on star topologies arranged in master-slave configurations [36,37,38]. Because such topologies are not scalable for supporting large numbers of IoT devices, recent BLE networks have focused on the use of Scatternets, which are hybrid star topologies, to enhance the network efficiency and scalability.

#### 3.2.1. Connection-Oriented Protocols

Scatternets typically form tree-structured networks, due to the master–slave relationship within a piconet (cluster), as well as the limited number of inter-cluster communication links between one BLE piconet and another. Consequently, scatternets adopt connection-oriented links for data transfer. This can lead to fragile network topologies, since the loss of inter-cluster links will result in disconnected clusters [39].

Mikhaylov and Tervonen [40] proposed an early BLE mesh solution, known as MultiHop Transfer Service (MHTS). The researchers utilized on-demand routing for multi-hop communication of nodes utilizing CC2540 SoCs developed by Texas instruments. In addition, the proposed solution worked well for 2-hop and 3-hop data forwarding. Nonetheless, further protocol enhancement is needed to support a higher number of nodes in a larger network due to various scalability issues.

Wang and Chiang [41] proposed a connection-oriented BLE-based tree topology protocol. The BLE-Tree Network utilizes the 37 Bluetooth data channels to form master-slave chains. Each device has two BLE interfaces, with one configured as a Master and the other as a Slave. Four processes, i.e., Master Agent, Slave Agent, Scan Center, and Sensor Center work together to maintain the tree topology. The Master Agent initiates connections to other devices (slaves) within its range. The Slave Agent broadcasts its UUID for discovery by the Master Agents of neighboring devices. The Scan Center is used by the Slave interface to listen for neighbor broadcasts and send messages to its associated Master Agent, while the Sensor Center tracks the MAC address of neighbors and transmit data to them. The achieved Packet Delivery Ratio (PDR) was much better than connectionless (broadcast) protocols.

Martinez et al. [42] utilized an existing mesh protocol to implement a mesh topology for doorbells in an office environment. The system used a Nordic Semiconductor board and the Softdevice libraries to demonstrate the efficiency of the BLE mesh network. Sirur et al. [43] proposed an on-demand routing technique with a weight-balancing approach for data communication optimization. Their system supports the dynamic organization of nodes for efficient data forwarding.

According to Balogh et al. [39], the use of scatternet topology in BLE 4.1 allows for long-range communications. In the paper, the authors proposed a service mediation concept that is based on the Named Data Networking (NDN) approach to overcome the limitations of the scatternet specifications in BLE 4.1. In contrast, Guo et al. [44] developed an on-demand multi-hop BLE routing protocol based on the BLE 4.1 scatternet topology. The proposed system was tested on real hardware, where the protocol performed well in terms of latency and resource utilization.

Bardoutsos et al. [45] proposed a multi-hop tree-based wireless network with multi-protocol gateways utilizing heterogeneous technologies, i.e., Wifi and BLE, to improve energy efficiency, and obtained promising results. Similarly, Dvinge et al. [46] measured the power consumption of nodes while using the FruityMesh protocol to show that connection-oriented Bluetooth mesh networks could be a suitable solution for off-grid applications due to its low power consumption. On the other hand, Murillo et al. [47] utilized a Software Defined Network (SDN) approach to achieve longer lifetimes for static BLE resource-constrained nodes. The proposed protocol was devised to balance between the number of connection events and energy consumption.

Ng and She [48] presented a novel BLE-based overlay mesh solution to address issues of best-effort scheduling (BES) and RSSI-based bounded flooding (RBF) to achieve mesh functionality in a BLE beacon-based network. Twenty BLE devices and four Android mobile devices were used to perform sensing/advertising and receiving tasks, with better performance as compared to existing solutions.

Jung et al. [10] introduced a BLE-based on-demand multi-hop routing protocol. The CbODRP protocol addressed topology configuration issues (node discovery, piconet, and scatter net formation) and cluster recovery procedures, using an on-demand (reactive) routing protocol. During node discovery, all BLE nodes send advertisements and update their neighbor count value to select a master device with the highest number of neighbors, using the device id as a tie-breaker. Subsequently, master nodes establishes piconets with its neighbors. Scatternet formation then occurs in two phases. In phase one, a neighbor node with the lowest id belonging to another piconet is selected as a relay node, and a link is established with that piconet. In phase two, slave nodes in the current piconet identify neighbors that belong to other piconets without established connections and inform the master of candidate relay nodes for new connections, as shown in Figure 3. For cluster recovery, the neighbor count metric is used to select a new master for a piconet in case of a master node failure. Routing overhead is reduced by grouping messages for batch transmission to master nodes and relay nodes. This reduces the energy consumption and route discovery delays compared to conventional on-demand routing protocols.

According to Darroudi and Gomez [49], connectivity is a vital aspect of any wireless mesh network. An analytical model for node connectivity probability was proposed. In their model, isolated nodes are considered to be outside the network. In addition, two nodes can only establish a connection with each other if they have an adequate number of time slots for communication. The formula for calculating the probability of at least one connection between a node and its neighbors, as well as the formula for the probability of node connectivity via k different paths were presented. The proposed model was validated via simulation and shown to be suitable for evaluating data-channel-based BLE mesh networks (DC-BMN).

#### 3.2.2. Connectionless Protocols

The use of connectionless fowarding in BLE Mesh topologies arose due to the limitations of the connection-oriented scatternet-based mesh protocols. Connectionless forwarding is implemented via the flooding of Broadcast packets throughout the BLE Mesh topology.

Murillo et al. [50,51] performed PDR, end-to-end delay, and power consumption measurements for both the Trickle (flooding-based connectionless) and FruityMesh (connection-oriented) protocols, respectively, using the Nordic nRF52 development board. It was found that the flooding approach achieved lower end-to-end delay at the expense of higher power consumption when compared to a connection-oriented approach for comparable PDR objectives.

According to Chiumento et al. [52], control parameters must be chosen very carefully in order to achieve a reliable and robust interconnected mesh network with the least congestion and packet loss probability. Furthermore, Hansen et al. [53] investigated the effect of relay node selection on the overall PDR of flooding-based BLE mesh networks. Better overall network performance was achieved while using fewer relay nodes; hence, optimal relay node selection is critical. Among three automated candidate relay selection algorithms, i.e., Greedy Connect, K2 Pruning, and Dominator, the K2 Pruning algorithm achieved the best performance at the expense of high data storage requirements. In contrast, the Greedy Connect algorithm is more efficient in terms of data storage requirements, while the Dominator algorithm enables fast network reconfiguration.

Finally, Li and Li [54] developed a Directional Ad-Hoc On-demand Multipath Distance Vector (D-AOMDV) protocol to address node mobility issues in a BLE mesh-based health monitoring system. The protocol addresses the link quality fluctuation issue of mobile BLE nodes using a Directional Link Quality Indicator.

#### 3.2.3. Real-Time Protocols

According to Leonardi et al. [11], the mechanisms for supporting real-time packet forwarding are undefined, since the BLE specification did not specify bounded packet delays. Bounded packet delays are necessary to meet real-time process deadlines. Patti et al. [55] proposed a BLE-based real-time multi-hop protocol, RT-BLE, with bounded message delays to overcome this limitation. RT-BLE tried to address the problem of transmission overlaps to support real time communication. The proposed protocol is capable of bounded latencies, but lacked coordinated transmission scheduling to avoid collisions.

The proposed MRT-BLE protocol [11] is an enhancement of the RT-BLE protocol [55] with the use of TDMA for collision avoidance. The network is subdivided into various sub-networks, each controlled by a master. Figure 4 illustrates the collision avoidance mechanism, where if MS1 is communicating with S1, S2, and S3 in one sub-network, then it will not accept any connection from M1 in the other sub-network until the previous communication is completed. Collisions are avoided by assigning different time slices (connection interval) to adjacent sub-networks. MS1, which belongs to both sub-networks, will maintain its connection with S1, S2, and S3 during its assigned time slice, but disconnect from them in order to connect to M1 in the other sub-network during the other time slice. Devices check for connection availability before transmission, while pending transmissions are queued until connections are reestablished. The drawback of this protocol is the need for offline network topology configuration, and the lack of support for mobile nodes.

#### 3.2.4. IPv6 Support

Luo et al. [56] proposed IPv6 over BLE (6LoBLE) to interconnect BLE networks with the Internet, in order to overcome the problem of limited communication distances. A neighbor discovery protocol for IPv6 over BLE mesh networks was proposed. In addition, the network structure and address auto-configuration/update processes of IPv6 over BLE mesh networks were also discussed. Performance analysis of neighbor discovery in 6LoBLE found that it was effective, even in a densely populated environment.

Moreover, Darroudi et al. [57] conducted a comparative study on the performance of BLE Mesh when compared to IPv6 BLE Mesh in terms of flexibility, message transmission, overheads, node density, and coverage area. IPv6 BLE Mesh was found to have various advantages as compared to BLE Mesh due to the inclusion of Internet connectivity support.

### 3.3. Heterogeneous BLE Mesh Networks

In this subsection, we will discuss proposed solutions which combine BLE Mesh networks with other wireless technologies to create heterogeneous BLE Mesh networks.

Ferranti et al. [58] developed a heterogeneous intelligent robotic network (HIRO-NET) to support post-disaster infrastructure-less communications. The proposed system consists of two layers: a short-range BLE Mesh-based communications layer is responsible for connecting survivors into local clusters, while the Very High Frequency (VHF) links in an overlay network layer interconnect autonomous robots into a metropolitan area network covering the disaster area. The VHF overlay network provides gateway access to interconnect the BLE Mesh clusters with each other. The results showed that the proposed network can be used to support disaster search-and-rescue operations in metropolitan areas.

Vijay et al. [59] proposed a heterogenous network monitoring system for air cargo, BLE-PLC, while using BLE and Power Line Communications (PLC) via power cables between the airport terminal and the aircraft. BLE Mesh technology was found to be better suited for air cargo monitoring, as it is self-contained, whereas competing technologies, such as RFID tags and camera-based visual inspection, depend on the use of properly positioned RFID readers [60] and availability of adequate lighting, respectively. The use of PLC technology for network backbone connectivity allows for the movement of BLE-enabled Gateways to the aircraft locations easily. The gateways are connected via PLC links to the monitoring network, to allow the tracking of cargo items equipped with BLE Mesh nodes in real-time during movement between the aircraft and storage facilities. The use of multi-hop BLE Mesh networking enables the tracking of cargo items located at a distance from the gateways. This is illustrated in Figure 5.

Garrido et al. [61] designed a heterogeneous wireless mesh network using LoRaWAN and BLE Mesh technologies to support Industry 4.0 requirements. The objective is to overcome the limited coverage provided by BLE Mesh networks, by using broadcaster and observer profiles to support connectionless communications among OperaBLE wearable BAN nodes equipped with various sensors. In addition, factory context information is collected using the LoRaWAN network for storage on servers in the GreenIS Factory network, and for dissemination via BLE Gateways to workers that are equipped with OperaBLE nodes, to provide alerts and other safety information.

Table 1 summarizes the selected protocols discussed in this section.

### 3.4. Research Direction for BLE Mesh Protocols

The coverage of recent BLE Mesh protocols highlighted the open research issues related to enhancement and scalability of BLE Mesh protocols. Because most BLE Mesh topologies are designed for the scatternet topologies using connection oriented communications, the robustness of BLE Mesh networks against node failure and mobility is limited [10]. Furthermore, the scalability of scatternets is hampered by the fact that only a limited number of inter-cluster links are used by most proposed protocols.

Multipath, connectionless BLE Mesh protocols for pure mesh topologies are needed in order to overcome the aforementioned limitations. However, the current proposed connectionless protocols mostly rely on broadcast-based flooding to forward packets. More efficient connectionless protocols that leverage directional forwarding (e.g., [54]) are needed to overcome the excessive packet forwarding overheads that are inherent in flooding-based solutions.

An orthorgonal issue is the support for real-time communications in BLE Mesh networks. While real-time packet forwarding is not the focus of this survey, scalable real-time communications in large BLE Mesh networks is not likely to be viable without the use of connectionless packet forwarding protocols, since bounded end-to-end delays [11] are difficult to achieve in connection-oriented scatternets.

There is a lack of support in BLE Mesh protocols for multi-hop and multicast packet forwarding, as the BLE specifications define broadcast and unicast transmissions. Multicasting provides an optimized approach for data delivery to a group of collaborating nodes. Such scenarios are often found in IoT applications, such as the control of a group of lights or the collection of sensor data for a Smart Home. Nonetheless, the proper discussion of multicast forwarding protocols is best done as a separate survey and is, therefore, not discussed in this paper.

Finally, further research into efficient distributed algorithms for mesh topology auto-configuration is needed for the bootstrapping of connectionless BLE Mesh networks.

## 4. BLE Mesh Security Issues

To understand the current state of the security challenges for BLE Mesh networks, it is necessary to first present an overview of Wireless Personal Area Network (WPAN) Networks attacks, in order to provide the context for categorizing BLE specific attacks and current BLE vulnerabilities. A summary of security features for BLE can be found in the Appendix B. Figure 6 provides an overview of the attacks affecting WPANs, as well as security threats that are specific to BLE.

### 4.1. Common WPAN Network Attacks.

#### 4.1.1. Breach of Confidentiality

Breach of Confidentiality involves the release of sensitive information to unauthorized recipients [62]. Strong encryption techniques and passwords, multiple level authentication techniques, etc. can be used to mitigate such breaches.

Password CrackingThis type of attack can be performed using brute force. It can be performed online as well as offline, but offline attacks are more dangerous, since attackers can conduct password cracking until they succeeded [63].Encryption AttacksThese attacks can target different network layers, i.e., physical, network, application layer, etc. In this kind of attack, the malicious node tries to decrypt encrypted data to obtain important information [63].Social Engineering AttacksThese attacks take advantage of interpersonal communications, where the attacker obtains important information, such as passwords via the gullibility of the target [63]. This can occur via the impersonation of a legitimate user to force a password reset for the target account. Similarly, passwords written and kept in exposed locations can also invite this kind of attacks.Packet SniffingPacket sniffing refers to capturing of network data packets during transit [64]. A packet sniffer is used to execute this attack.

#### 4.1.2. Breach of Integrity/Authentication

Client and Server Authentication is considered to be the first layer of network protection [65]. It is to ensure end-to-end accuracy, trustworthiness, and validity of data transfers [66]. Moreover, some of the major attacks on network integrity [67] are highlighted below.

Eavesdropping AttacksEavesdropping attacks can be passive or active. An attacker quietly monitors message transmission and gathers useful information for the desired purpose in passive attacks [62]. In contrast, active attacks occur when fraudulent nodes participate in communications, posing as legitimate nodes to obtain important information for misuse.Man in the Middle (MITM) AttacksIn this type of attack, a malicious node inserts itself into the communications channel between two legitimate nodes, while maintaining the facade that they are communicating with each other directly [63]. In the case of BLE, both legitimate GAP central and peripheral nodes will be associated with the impostor node to monitor the messages between the two legitimate nodes.Impersonation AttacksThese assaults can be as simple as fake emails authorizing a malicious person as an authorized user to obtain credentials for system access [62].Mac Spoofing AttacksMedia Access Control (MAC) is a form of impersonation attack, where the hard-coded address of the Network Interface Card (NIC) is modified to that for a legitimate device [62,67]. This causes network equipment accept connections from, or to deliver data to the malicious device.Replay AttacksA unique, genuine message is retransmitted, or it is delivered late to the destination, influencing the efficiency and operation of the system [62].Relay AttacksRelay attack is similar to MITM Attacks. A malevolent node inserts itself into the communications channel between two nodes and forwards copied information to another node for illegitimate use, without the original nodes being aware of the information leak [67].

#### 4.1.3. Breach of Availability

Data services should always be available and accessible to authorized users [64]. Common attacks that contravene data availability [68] are presented below.

Physical AttacksPhysical attack refers to breaching of physical security protections, such as theft. It also includes the disruption of wireless communication channels, such as the use of radio frequency jamming attacks. In addition, device cloning, board pin-jacking, and device tampering attacks come under the umbrella of physical attacks [69].Battery Exhaustion AttacksWPAN devices are usually battery operated devices that enter sleep mode when inactive. Battery Exhaustion Attacks force continuous fraudulent connection requests to drain the battery and cause the device to become unavailable.Denial of Service (DoS) AttacksDoS Attacks occur when an attacker floods the device with continuous connection requests that consume an enormous amount of network bandwidth [70]. DoS occurs when legitimate requests are not serviced due to overload on the device.BotnetsBotnets are a collection of malware-compromised Internet-connected devices which enable hackers to take control of the devices [63]. An intruder usually takes advantage of a botnet to instigate the botnet attacks that result in credential and information leaks, unauthorized access, DoS attacks, etc.

Current research into IoT device security mostly focused on overcoming network-based attacks by addressing weaknesses in Integrity and Authentication verification protocols. Lai et al. proposed CPAL [71], a secure roaming scheme for machine-to-machine communications that can partly mitigate MITM attacks. In addition, Lai et al. later developed GLARM [72], a secured authentication system for machine-to-machine communications that can efficiently mitigate MITM attacks.

Chen et al. [73] proposed S2M, an acoustic fingerprint-based authentication protocol that can deter against MITM and Replay attacks for wireless devices. Chuang et al. [74] proposed a lightweight authentication protocol for IoT devices, which has the capability to do static as well as continuous and dynamic authentication. Moreover, it can protect against MITM, impersonation, eavesdropping, and replay attacks. Consequently, the proposed protocol by Chuang et al. [74] appears to be suited for enhancing authentication features that are found in BLE nodes due to its lightweight design.

### 4.2. BLE Specific Network Attacks

BLE devices are resource-constrained and therefore susceptible to additional attacks, which may not be easily exploited against resource-rich devices, according to Santos et al. [75]. Moreover, there are various new attacks that force key renegotiation in paired devices, one of the fundamental security mechanisms used in BLE. For example, the BLE Injection-Free attack developed by Santos et al., successfully forced Long Term Key (LTK) renegotiation for a connection despite unsuccessful attempts via other attack methods. Figure 6 summarizes this and other BLE specific attacks [67].

Key Negotiation of Bluetooth (KNOB) AttackIn this kind of attack, an attacker without any prior knowledge of any encryption key or link can make two or more victims agree on an encryption key. Hence, an attacker might brute force the encryption key, decrypt ciphertext for eavesdropping, which then enables an attacker to send a message as a legitimate user. Additionally, all Bluetooth versions are susceptible to this attack [76].BLE Injection Free AttackThis attack uses MITM attacks as a vector to cause DoS in a BLE network [75].Bluejacking AttacksIn this attack, a foe starts an assault by sending unsolicited data to the target user. Successful attacks insert fraudulent contacts into the victim’s address book, similar to how portable phishing and spam attacks function [63].BluebuggingBluebugging attack allows for an attacker to access the victim’s cell phone commands and takes over the phone and short message service subsystems [63]. In this attack, the intruder can modify a list of contacts and record phone calls by eavesdropping on the call.BluesnarfingIn this assault, an attacker obtains unauthorized access to a Bluetooth enabled device and steals information. This attack not only breaches authentication, confidentiality, availability, but also does not leave any fingerprints for auditing and forensic purposes [67].BluebumpThis attack takes advantage of the weak Bluetooth link key-handling protocol, thus allowing an unauthorized device to access services as a legitimate user. Bump attacks can cause data theft and the manipulation of mobile internet connectivity services [67].BluedumpBlueDump is an attack where the attacker cause a Bluetooth device to abandon its link key and pair with the attacker’s Bluetooth device instead [67].BluemackIt works like the Bluetooth Denial Of Service (DoS) assault, where a Bluetooth-equipped device is compromised via malformed requests from an attacker. The device becomes unresponsive and eventually stops working due to battery exhaustion [67].BluechopThe attacker uses an unassociated device to cause an existing slave node to disassociate from the master node to disrupt the operation of the piconet [67].

### 4.3. Current BLE Vulnerabilities

Because new vulnerabilities are continually being discovered and existing implementation-related vulnerabilities (which are not due to flaws in the protocol design) are fixed by vendors, it is not possible to provide a definitive list of BLE vulnerabilities. A recent report identified various Bluetooth security susceptibilities, collectively known as the SweynTooth exploits, resulting in DoS attacks on affected devices [12]. Figure 7 summarizes these exploits.

### 4.4. Bluetooth Security Enhancements

In view of the various Bluetooth security threats, efforts to improve the BLE security focused on enhancing authentication and improving the integrity of the scatternet formation process.

Yu and Wang [77] proposed a security enhancement for the Bluetooth Topology Construction Protocol (BTCP) to secure scatternet formation. The protocol can overcome MITM attacks by ensuring proper authentication during the scatternet formation process. Similarly, Sadghzadh et al. [78] proposed a security protocol for Bluetooth networks by combining encrypted key exchange protocols with a special focus on authentication.

Xu and Yu [79] developed an enhanced Bluetooth pairing protocol that can mitigate MITM attacks. Existing solutions using Elliptic-curve Diffie–Hellman (ECDH) key agreement are strong against passive eavesdropping attacks, but cannot mitigate MITM attacks. The proposed protocol uses a strong public key exchange mechanism to protect against MITM, passive eavesdropping, replay, and impersonation attacks. Similarly, Diallo and Wajdi [80] developed a protocol that implements double layered encryption to secure data communications. The proposed protocol avoids the weaknesses of Secure and Simple Pairing (SSP) by eliminating cleartext public key and password exchanges, and using Hash-based Message Authentication Code (HMAC) to prevent message tampering during the key exchange process.

According to Fan et al. [81], Bluetooth version 4 defines a Numeric Comparison (NC) pairing method that requires both input and output capabilities. They studied the shortcomings of a pin-based authentication model for a scenario in which one of two devices does not have an output capability.

Priyanka and Nagajayanthi [82] developed a link-layer security mechanism for the Bluetooth stack to perform message authentication and integrity, prevent MITM attacks, and detect message alterations.

Nai and Yohan [83] designed a lightweight Physical Unclonable Function (PUF)-based authentication protocol for joint authentication and maintenance of secrecy for the session key. The proposed protocol was used to implement a micropayment system, where users can securely perform transactions via BLE-enabled wearable devices. Subsequently, the authors verified that the developed system is capable of preventing session hijacking and bogus payment attacks, as well as traditional attacks, such as passive eavesdropping, replay, MITM, and impersonation attacks.

In contrast to the use of challenge-response-based algorithms, Cha et al. [84] proposed a blockchain-enabled IoT gateway for BLE-based devices, in order to preserve user privacy and enforce user preferences when accessing IoT devices. Ethereum blockchain-based smart contracts are generated between the device and the IoT gateway to specify device information and device privacy policies. Users wishing to access the services provided by the IoT devices can specify their usage preferences as smart-contracts between the user and the blockchain-enabled IoT gateway. The gateway mediates transactions between the users and IoT devices, in order to protect the privacy of the users based on their specified usage preferences via the specified smart contracts.

Table 2 is a summary of the Bluetooth security enhancements.

### 4.5. Relevance to BLE Mesh Networks

Even though enhanced security features have been introduced in Bluetooth 4.2 and Bluetooth 5, BLE mesh networks are still vulnerable to security breaches. Authentication and Integrity protocols are mostly designed for bipartite communication flows. Because most of the proposals involve securing the pairing process between adjacent BLE nodes in a piconet or scatternet, additional research is needed to study the applicability and effectiveness of these algorithms for multihop BLE Mesh networks involving a large number of nodes.

## 5. IDS for Related WSN Technologies

Because zero-day attacks are an ever present threat against interconnected IoT networks, it is necessary to adopt intrusion detection techniques to provide early warning against such threats. Due to the fact that research into Instursion Detection Systems (IDS) for BLE Mesh Networks is still in its infancy, a survey of techniques used by IDS for other more established wireless technologies, as shown in Figure 8, is expected to provide direction for further research in this area.

An IDS is a hardware or software module that can monitor the network for any suspicious activity. If irregular activities were detected, the IDS will report the anomaly to the administrator or network operation centre (NOC) to take necessary action in response.

### 5.1. IDS for IoT Networks

In this paper, IoT Networks refer to networks that are less resource constrained when compared to WPAN-based networks. Typically, IoT devices use Ethernet or Wi-Fi technology for communications, and may also be mains-powered. Hence, they are not subject to the resource limitations that are experienced by battery-powered devices.

Prabavathy et al. [85] devised an IDS based on fog computing while using an Online Sequential-Extreme Learning Machine (OS-ELM) algorithm to efficiently detect attacks in a large scale network of IoT-based devices. The use of distributed intelligence in local fog nodes resulted in a more efficient, flexible, interoperable system with 25% faster attack detection capability as compared with centralized approaches.

Choudhary and Kesswani [86] designed the Key-Match (KMA) and Cluster-Based (CBA) IDS and IPS algorithms to defend against sinkhole and selective forwarding routing attacks, with high true positive detection rates. Tian et al. [87] proposed an anomaly-based intrusion detection scheme by merging a deep-learning method (DLM) with a shallow-learning approach (SLA). In the proposed framework, the DLM uses a deep auto-encoder for feature learning, whereas the SLA consist of a Support Vector Machine (SVM) implementing an Artificial Bee Colony (ABC) algorithm parameters optimization while using Five-fold Cross Validation (5FCV). The proposed framework performed better than the Principal Component Analysis (PCA)-based approach.

DoS attacks represent one of the main source of attacks on IoT infrastructures. According to Jan et al. [88], the presented IDS techniques cannot fully prevent intrusions. The authors proposed a very lightweight Machine Learning (ML)-based approach for the detection of unwanted data injection by attackers. The received data attributes were used as inputs to the SVM classifier, to achieve effective intrusion detection in IoT networks. SVM was found to be more accurate when compared with competing Neural Network (NN), k-Nearest Neighbor (k-NN), and Device Tree (DT) approaches.

Anthi et al. [89] developed a three-layer IDS that monitors the normal behavior of each IoT device in the network, identify malicious packet transmission, and performs attack detection while using a supervised machine learning approach. The proposed system was evaluated using twelve types of attacks from four categories of network exploits, i.e., DOS, MITM, reconnaissance, and replay attacks, in a smart home testbed comprising of eight IoT devices.

Zolanvari et al. [90] analyzed the most common protocols utilized by Supervisory Control And Data Acquisition (SCADA) Industrial Internet of Things (IIoT) devices for security vulnerabilities. The use of machine learning techniques to protect against attacks, such as backdoors, command, and SQL injection attacks were studied using a testbed.

Liang et al. [91] analyzed and discussed the trade-offs inherent in the use of machine learning in general, as well as from the security perspective. The use of machine learning has improved IDS detection rates for IoT. However, there are inherent weaknesses in the machine learning approach, where the training datasets used for machine learning can be subverted to facilitate attacks against the IoT infrastruture.

In contrast, You et al. [92] concluded that rule-based monitoring solutions for IoT intrusion detection outperformed anomaly-based detection approaches. Sharma et al. [93] designed a Behavior Rule-based intrusion detection methodology (BRIoT) for mission-critical cyber-physical systems. Based on the operation profile of IoT device, BRIoT will automatically generate a set of requirements and behavior rules, verify the generated rules, and convert them into a state machine for runtime intrusion detection.

Alhakami et al. [94] proposed a Bayesian-based approach for IDS while using the Infinite Bounded Generalized Gaussian mixture (InBGG) model. The model incorporated a feature selection mechanism to remove extraneous features that can compromise the model’s accuracy and efficiency. A comparative analysis using different data sets (KDDCup’99, KYOTO 2006+, and ISCX) demonstrated the effectiveness of the InBGG model in terms of a better False Positive Rate (FPR) and Accuracy results.

Arshad and Azad [95] proposed an IDS combining host and network-based frameworks for more efficient and accurate detection capabilities. The proposed system can detect multi-staged attacks on the IoT infrastructure via its combined approach. Various attack scenarios were evaluated while using the Contiki OS simulator (Cooja) with good performance and effectiveness.

Abhishek et al. [96] proposed a Generalized Likelihood Ratio Test (GLRT) for detecting attacks on communication links between IoT devices and Access Points, by examining the uplink unicast packet retransmission ratios that arise from relay-node packet drops, and downlink and broadcast packet drop ratios experienced by the IoT devices. The proposed approach was able to reduce the number of false alarms and missed detections.

According to Choi et al. [97], IDS systems for the IoT environment based on pattern matching and behavior-based statistical methods were less effective when compared to their proposed onotology reasoning approach for access control. Their model defined context inference rules for a cloud-based IoT security provisioning framework, evaluated in a smart meter-based power delivery system scenario. The proposed system was able to achieve high intrusion detection rates while using the developed inference rules.

In contrast, Nguyen et al. [98] proposed the use of software-defined networking (SDN), network function virtualization (NFV), and machine learning-based detection for implementing IDS solutions, due to their performance advantages. SeArch, a Network-based Intrusion Detection System for a SDN-based cloud IoT environment, comprises of tiered layers of nodes that collaborate to detect irregularities and devise mitigation policies for the IoT gateway devices to stop intrusions.

Table 3 summarizes the various IDS proposals for IoT Networks.

### 5.2. IDS for ZigBee-Based Networks

Zigbee is a commonly used WPAN Mesh protocol due to its shared-access radio channel model that is based on the IEEE 802.15.4 standard. Because Zigbee devices are often battery-powered, it experiences various resource constraints. Zigbee is a competing technology to BLE in terms of coverage and use cases. This section examines the various IDS solutions proposed for Zigbee Mesh Networks.

Jokar et al. [99] highlighted the vulnerability of Home Area Networks (HAN) that participate in a smart grid environment. The expected behavior of IEEE 802.15.4 devices was extracted from the standard as specifications, in order to enable the proposed IDS to detect anomalies due to physical and MAC layer attacks against Zigbee-based networks. The system was shown to provide good protection against known and unknown anomalous events and attacks. Similarly, Stelte and Rodosek [100] proposed an anomaly-based IDS to protect ZigBee-based WSN nodes against KillerBee supported threats.

Anomaly-based detection was also adopted by Baalbaki et al. [101] for their proposed ZigBee IDS. The system was able to detect different attacks, such as DoS, flooding, and pulse DoS, with a high detection rate and low false positive rate, when compared with signature and specification-based IDS approaches. In contrast, Maphatsoe and Masinde [102] used fuzzy logic reasoning methods in their anomaly detection algorithm to achieve better efficiency in detecting flooding attacks during the node discovery and association processes, in comparison to existing solutions.

Jokar et al. [103] combined a model-based IDS with a machine learning-based IPS into their proposed security system for ZigBee-based HANs. Their proposed system utilized both the Smart Energy Profile 2 and the IEEE 802.15.4 specifications to define the expected behavior profiles for their models.

Wormhole attacks are a major threat to WSNs due to its multi-hop nature, according to Jegan and Samundiswary [104]. The authors proposed the use of an optimized trust-based watchdog mechanism in their Energy Efficient IDS (EE-IDS) for Zigbee-based networks. The proposed system was evaluated for static and mobile Zigbee networks in terms of its effectiveness in detecting wormhole attacks. The EE-IDS module was further augmented with the Energy-Efficient IDS with Energy Prediction (EE-IDSEP) module [105] to protect against wormhole and DDoS attacks. The system was able to detect wormhole attacks against Ad-hoc On-Demand Distance Vector (AODV), Shortcut Tree Routing (STR), and Opportunistic Shortcut Tree Routing (OSTR) protocols, while the DDoS attack detection performance was better than the performance of Energy Efficient Trust System (EE-TS).

Chen et al. [106] noted that Low-rate DoS (LDoS) attacks are new types of WSN attacks that are difficult to detect. The authors proposed effective algorithms for LDoS detection in Zigbee WSN by combining Hilbert–Huang Transforms (HHT) with Trust Evaluation (TE) approaches. Therefore, the algorithms are suitable for incorporation into IDS for detecting such attacks.

Table 4 summarizes the various IDS proposals for Zigbee Networks.

### 5.3. IDS for 6LoWPAN-Based Networks

6LoWPAN is a protocol stack developed by the Internet Engineering Task Force (IETF) to support IPv6 packet transmissions over WPANs. While there is intersection between 6LoWPAN networks and other non-IPv6 WPAN networks, IPv6 support introduces additional issues that merit a separate discussion section.

Kasinathan et al. [107] designed an IDS for 6LoWPAN for the ebbits network framework based on Suricata to detect DoS attacks. The proposed Network-based IDS runs on a host computer to collect data from various IDS Probes installed in the 6LoWPAN network, to avoid overloading 6LoWPAN devices with limited processing capabilities.

Cervantes et al. [108] proposed intrusion detection of sinkhole attacks on 6LoWPAN for internet of things (INTI), an IDS for detecting sinkhole attacks on 6LoWPAN routing services. The system incorporated watchdog, reputation, and trust metrics in order to characterize the behavior of devices, thereby improving the attack detection rate while reducing the false positives and false negatives.

However, Surendar and Umamakeswari [109] noted that techniques used by SVELTE and INTI require significant resources and experienced high packet drop ratios. Their proposed Intrusion Detection and Response System (InDRES) utilized a constraint-based specification model in order to provide better sinkhole attack detection with less overhead and energy consumption.

Napiah et al. [110] proposed the Compression Header Analyzer Intrusion Detection System (CHA-IDS), which utilized both anomaly and signature-based approaches to verify 6LoWPAN compression header data, to protect against 6LoWPAN routing attacks. The Machine-learning classifier used by CHA-IDS was trained while using the 6LoWPAN compressed headers to avoid header decompression overheads.

Althubaity et al. [111] proposed an Authenticated Rank and routing Metric (ARM), a hybrid specification-based IDS, in order to protect against routing attacks against RPL (Routing Protocol for Low-power and lossy networks) used in a IPv6 over Time-Slotted Channel Hopping (6TiSCH) network. The 6TiSCH standard defines the transport of 6LoWPAN traffic over an IEEE 802.15.4e WPAN, and it is susceptible to RPL topology attacks. The proposed ARM IDS was able to prevent fraudulent nodes from inserting themselves into the RPL topology, while requiring low overhead.

Farzaneh et al. [112] proposed a distributed lightweight anomaly-based IDS using the standard deviation of received RPL control message time intervals as threshold values to facilitate anomaly detection. The threshold-based approach consumes very little overhead and can, therefore, be deployed in large 6LoWPAN networks.

Nonetheless, most of the available IDS solutions can detect attacks, but cannot quantify the attack severity [113]. Ramos et al. proposed the Node Security Quantification (NSQ) probabilistic model to quantitatively evaluate the severity of an attack on affected networks. The attack severity is quantified while using the Message Security Value (MSV) and the Damage Level (DL) to determine the impact of compromised nodes on the integrity of data sent via the network.

Table 5 summarizes the features of IDS solutions for the 6LoWPAN environment.

### 5.4. IDS for BLE Networks

In this section, we have surveyed the latest available works that are related to IDS in BLE-based networks.

Early work on Bluetooth IDS by OConnor and Reeves [114] used a Network-based IDS to monitor Bluetooth communications traffic in nearby devices in order to detect malicious behavior, by utilizing a pattern matching approach. The weakness of the proposed system was that it could only detect known attacks and is therefore unable to deal with zero-day attacks.

Guo et al. [115] proposed threshold-based IDS and IPS algorithm to mitigate battery exhaustion attacks in BLE networks. Malicious nodes are blacklisted in order to prevent them from accessing the BLE scatternet. The proposed method was able to reduce the impact of such attacks and substantially increase the network lifetime.

Mateusz and Michal [116] proposed a anomaly-based IDS for BLE Mesh nodes. A machine-learning traffic classification algorithm was used to detect malevolent behavior, while using distributed watchdogs for cooperative attack detection. Simulation and testbed results confirmed the importance of watchdog location selection to ensure the effectiveness of the IDS.

Table 6 is a summary of BLE-based IDS research to date.

### 5.5. Relevant Approaches for BLE Mesh IDS

Various competing IDS approaches, including anomaly detection using machine-learning techniques, rule and specification-based methods, as well as statistical threshold and behavior-reasoning approaches, have been proposed for other wireless environments. Nonetheless, IDS approaches for BLE Mesh networks must be inherently lightweight and distributed, in order to cater for the limited-resource environment experienced by a large population of networked BLE nodes. Distributed IDS systems [85] would also be less taxing on the BLE Mesh links, since less data would need to be sent across the network to facilitate intrusion detection, as compared to a centralized approach.

The use of probe devices [107] and watchdogs [104,116] may be one way to overcome the resource constraints but that can be adopted only when the locations of the BLE Mesh nodes are relatively stable, since poor probe or watchdog placement strategies will greatly affect the IDS performance [116].

Finally, in addition to protecting the integrity of the network, protecting the integrity of the information carried by the system [113] in spite of successful intrusions should also be explored as an objective of future IDS.

## 6. Conclusions

BLE Mesh is an emerging wireless mesh technology that is built upon the Bluetooth Low Energy standard. Nonetheless, most BLE Mesh topologies are overlay networks constructed using the connection-oriented scatternet links defined by the legacy Bluetooth architecture. The first part of this survey explored various implementations of hybrid BLE Mesh networks based on scatternets. BLE pure mesh networks, where each node acts as a forwarding peer, rely on the use of connectionless flooding and they incur a lot of transmission overheads. Research into optimal non-flooding-based connectionless BLE Mesh protocols remain an open research problem. Protocols for the efficient transmission of multicast data in BLE Mesh networks are also lacking. Moreover, there is a need for efficient auto-configuration mechanisms to support bootstrapping of BLE pure mesh networks.

Additionally, energy efficiency is an important design criteria for BLE Mesh networks, especially due to their dependency on power-constrained devices. From the analysis of the research papers presented in Table 1, there is a need to evaluate the performance of BLE Mesh protocols with respect to energy efficiency as a main focus. This is necessary for the enhancement of existing protocols and the development of new energy efficient mesh communication protocols for BLE that can support existing and new distributed sensing effectively.

Security issues for BLE and BLE Mesh networks were studied in the second part of this survey. Despite the additional security measures that are introduced in Bluetooth 4.2 and Bluetooth 5, the focus of these security measures were mostly on securing the communications channel between a pair of nodes. Efficient techniques for performing authentication and ensuring the integrity of the mesh network are needed to support the widespread deployment of BLE Mesh networks. Furthermore, IDS are needed to augment the available security mechanisms and to protect the BLE Mesh network from zero-day attacks.

Viable IDS solutions for BLE Mesh networks remain a nascent research area. Consequently, a comparative survey of IDS approaches for related low-power wireless protocols was used to map out potential approaches for enhancing IDS for BLE Mesh networks. Light-weight, distributed IDS approaches are necessary for BLE Mesh networks due to the resource limitations of battery-powered BLE nodes. Although the use of IDS probe devices and watchdogs could potentially alleviate the resource constraints, their effectiveness is highly dependent on the optimal placement of such devices within the network.

Finally, securing the BLE Mesh network should include not only ensuring network integrity, but also ensuring information integrity, since BLE Mesh networks are often used for forwarding data in IoT applications.

In short, energy efficient BLE pure mesh protocol capable of multicasting and topology auto-configuration is needed to support scalable applications effectively. In addition, intelligent IDS that can protect the BLE mesh networks from attacks are also crucial for the secure operation of these applications.

## Figures and Tables

**Figure 1 sensors-20-03590-f001:**
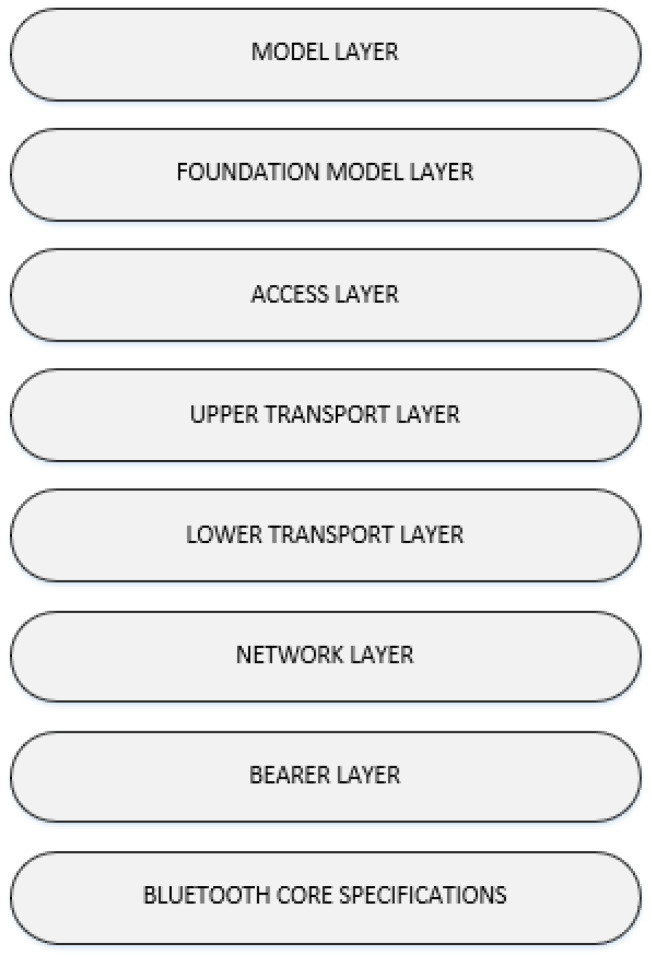
Bluetooth Low Energy (BLE) Mesh System Architecture [17].

**Figure 2 sensors-20-03590-f002:**
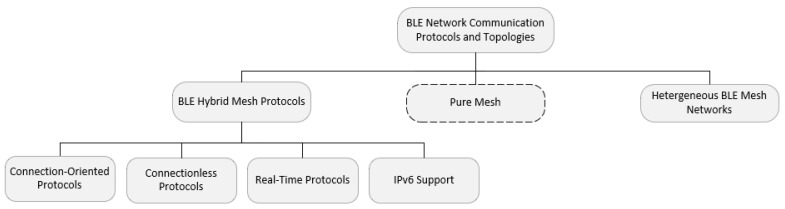
Classification of BLE Communication Protocols.

**Figure 3 sensors-20-03590-f003:**
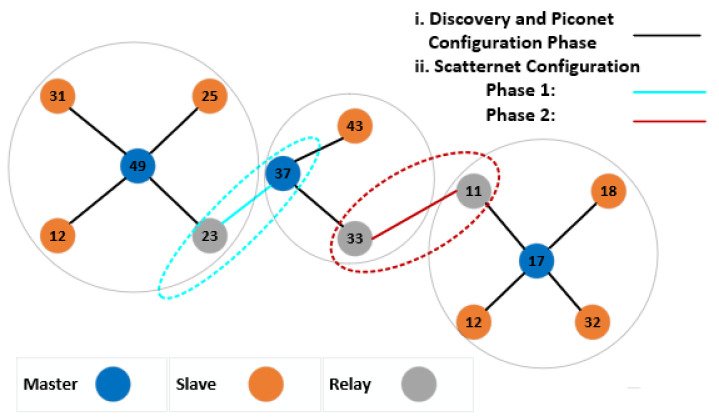
Cluster-based On Demand Routing Protocol [10].

**Figure 4 sensors-20-03590-f004:**
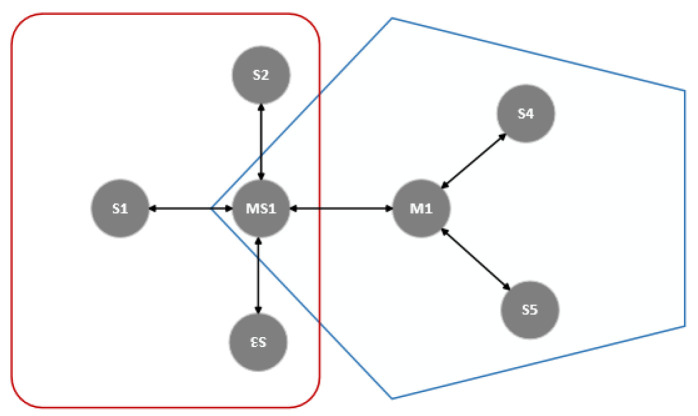
MRT-BLE [11].

**Figure 5 sensors-20-03590-f005:**
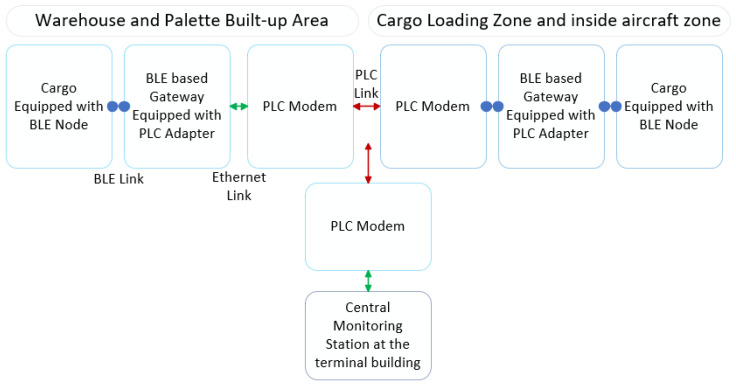
BLE MN Smart Cargo [59].

**Figure 6 sensors-20-03590-f006:**
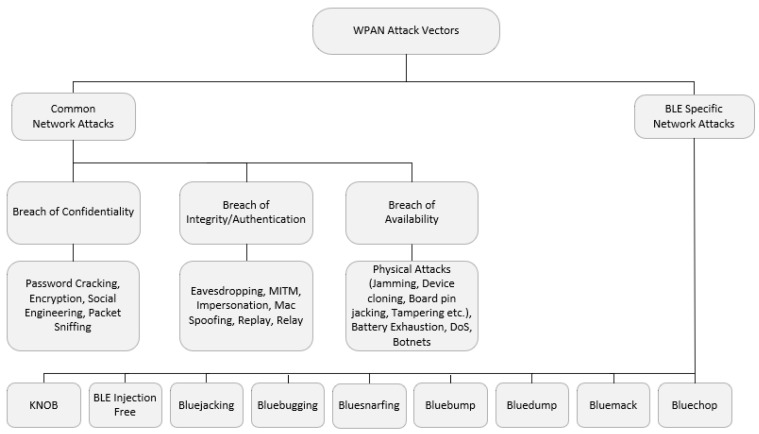
Wireless Personal Area Network (WPAN) Attack Vectors.

**Figure 7 sensors-20-03590-f007:**
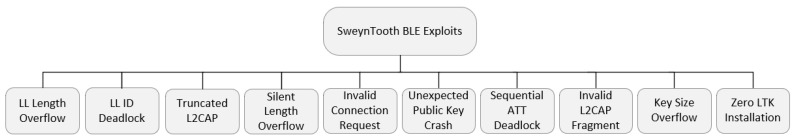
SweynTooth BLE Exploits.

**Figure 8 sensors-20-03590-f008:**
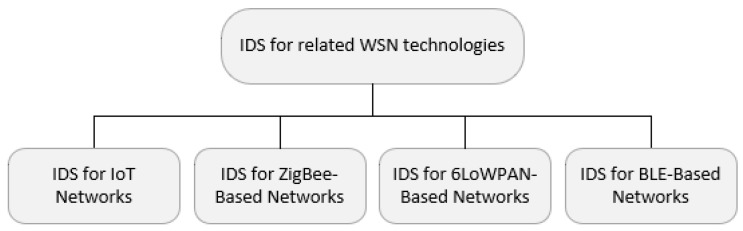
Intrusion Detection Systems (IDS) for related WSN technologies.

**Table 1 sensors-20-03590-t001:** Summary of various BLE Mesh Protocols.

Ref.	–Protocol –Homogeneous BLE or Heterogeneous	-Connection-Oriented (C) /Routing (R) -Flooding (F)/(R)	Test Bed (T)/ Simulation (S)	PDR	End to End Delay	Power Consumption	Nodes Other Measurements (OM) Throughput	Pros and Cons
[10]	–CbODRP –Homogeneous BLE	C/R-Reactive	S	×	-Route discovery delay 40–100 ms with 50–90 nodes	Approx 250–500 mA with 50–90 nodes	50 to 90 **Nodes** **OM**-Route Req Messages 0–20 with number of nodes 50–90 -Control Paket Overhead 12–88 with Route Discovery interval 1–10 s	**Pros**-Good contribution towards hybrid mesh protocols **Cons**-No new pure mesh protocol proposed -Lack of Hardware Implementation
[11]	–MRT-BLE –Homogeneous BLE Network (Static Nodes) (Bounded Packet Delays in Mesh Network)	C (Static Routing Configure Offline to get bounded delays)	T-(BLE) X-NUCLEO-IDB05A1	-Single Hop (100%) -Two Hops (Approx 100%) -Three Hops (96%) -Four Hops (97%)	-Single Hop (120 ms) -Two Hops (390 ms) -Three Hops (810 ms) -Four Hops (1050 ms) -Five Hops (1400 ms)	×	8 Nodes	**Pros**-Positive contribution (hybrid mesh protocols) **Cons**-Require Dynamıc Configuration Mechanism for free movement of nodes -Missing Dynamic topology management mechanism -Need Realtime routing for efficiency
[41]	–BLE-Tree Network –Homogeneous BLE	C and R (Reactive)	T Raspberry Pi 3 Model B (BLE 4.1) S	100% (for 2 p/s) 97.5% (for 5 p/s) 82% (for 10 p/s)	Round-Trip Time: For 1 Hop = 100 ms 2 Hops = 200 ms 3 Hops = 250 ms 4 Hops = 340 ms 5 Hops=360 ms 6 Hops=530 ms	×	40 Nodes	**Pros**-Efficient Breadth First Search algorithm **Cons**-No detailed analysis (other tree-based protocols)-Security (Authentication methods not discussed)
[46]	–FruityMesh –Homogeneous BLE	C (FruityMesh (FM)) /R-Reactive	T-Nordic Thingy: 52 IOT sensor kit -nRF52DK -nRF6707	×	×	Note: Connection Interval (CI) (7.5–400 ms) (a) 0.65–0.1 mA (Adv Interval 100 ms with CI) (b) 0.6–0.03 mA (Adv Interval 600 ms with CI) (c) Network Life 10–250 days with CI	1 to 3 **Nodes** **OM** with CI from 7.5–400 ms and advertising interval (100 and 600) the current drain is 0–0.65 mA **Throughput** Approx from 8–0.5 kB/s for CI 5–400 ms and max 3 packets/interval -with CI > 400 ms is 150 b/s	**Pros**-Good contribution related to power consumption and current drainage **Cons**-Need to work on more number of nodes and further analysis of performance measurements
[49]	–DC-BMN –Homogeneous BLE	C	S-Matlab	×	×	×	100 **Nodes** **OM** For N Slot 10	**Pros**-Analystic Model for node isolation probability **Cons**-No Performance Measurement-No Testbed
[50]	–FruityMesh and Trickle –Homogeneous BLE	C (FM) /R-Reactive F-(Trickle (TR))	T-nRF52 (BLE 5)	FM: 40% (10 p/s) TR: 38%(10 p/s)	FM: Approx 3.8 s TR: 0.35 s	FM: 9.4 mW TR: 28.5 mW	7 Nodes	**Pros**-Author performed good comparison between C and F networks **Cons**-No new algorithm or method proposed
[51]	–FruityMesh and Trickle –Homogeneous BLE	C (FM)/ R-Reactive F-TR	T-nRF52 (BLE 5) -Five Hardkernel Odroid-C2 -Netgear GS108T 8-port switch	-FM: 100%, 90%, 40% with 1, 5, 10 p/s resp -TR: 100%, 80%, 38% with 1, 5, 10 p/s resp	-FM: 0.3, 3.7, 3.9 s with 1, 5, 10 p/s resp -TR: 0.4, 0.3, 0.3 s with 1, 5, 10 p/s	FM: 9 mW TR: 28 mW	37 Nodes	**Pros**-Good implementation of available hybrid mesh protocols
[52]	–BLE Mesh –Homogeneous BLE	C/R-Reactive	T- nRF52 (BLE 5)	High	Low	×	12 Nodes	**Pros**-Used network inference for determination of node settings and design choices **Cons**-Require more experimentation
[53]	–K2 Pruning Greedy Connect and Dominator –Homogeneous BLE	F-(Greedy Connect (GC) K2 Pruning (KP) Dominator(D))	S-Matlab	-Area (330 × 330 msq) K2: Approx 80%–8% with packets 5–200 p/s GC:65%–8% with packets 5–200 p/s	×	×	1000 Nodes	**Pros**-Comparative study of various flooding techniques **Cons**-No proposed protocol
[54]	–D-AOMDV –Homogeneous BLE	F	S-Matlab	Approx 75%–88% with number of nodes 10–40	×	×	40 Nodes	**Pros**-Good Simulation Results with 40 Nodes **Cons**-Require Analysis of more performance parameters
[55]	–RT-BLE –Homogeneous BLE (Static Nodes) (Bounded Packet Delays in Mesh Network)	C (Static Routing Configure Offline to get bounded delays)	T-(BLE) X-NUCLEO-IDB05A1	×	20 ms	×	4 Nodes	**Pros**-Positive contribution (hybrid mesh protocols) **Cons**-Require Dynamıc Configuration Mechanism for free movement of nodes -Require Dynamic topology management mechanism -Need Realtime routing for efficiency
[59]	–BLE-PLC –Heterogenous	C/R-Reactive	T-nRF52832 (BLE 5) -PLC Modem IEEE 1901 -Wiznet 5550	-BLE Node Distance 0.6 m, transmit power −4 to 4 dBm a. 93%–97.5% (without wifi interference) b. 91%–96.5% (with wifi)	-BLE Node Distance 0.6 m, transmit power (−4 to 4 dBm) a. 8.2–6.3 ms (without wifi) b. 8.4–6.4 ms (with wifi)	×	12 Nodes	**Pros**-Good efficient smart cargo **Cons**-Tested only for PDR and Delay
[61]	–OperaBLE –Heterogenous (BLE, LoRaWAN, BAN) (Mobile Nodes)	C/R-Reactive F	T-Arduino UNO -BLE112 and CSRmesh -Light Blue Bean -Raspberry Pi 2 B -LoRaWAN etc	**BLE Mesh** -Supervisor Request/Response 100% and 92% -Supervisor Taps (96%) -Movements Tx: (Approx 96%) **OperaBLE** -Supervisor Taps (92%) -OperaBLE movement (Approx 90%)	**BLE Mesh** -Requests (Average Delay 0.347 s) -Taps (1.407 s) -Movement (Average time per packet 0.290s)	**OperaBLE** -Program Running 14 mA -Sleep Mode 2 mA	**Nodes** Not Mentioned **OM** Security at work -Heart rate measure (Error rate while fatigued 6.5% relaxed 1.05% working, 2.9%)	**Pros**-Good Heterogenous network for the industry **Cons**-Require development of coginitive systems for intelligent support enhancement -ML techniques can be applied for movements -Security can be improved

**Table 2 sensors-20-03590-t002:** Enhancements for Bluetooth Security.

Ref.	Protocol	Attack Manipulation
[77]	BTCP	-Protocol to Secure Scatternet Formation -Efficient Against MITM Attack
[78]	RAP-BE	-Combined encrypted key exchange protocols -Efficient Against DoS and Relay Attacks
[79]	SEBSPP	-Utilized strong public key exchange mechanism to Mitigate MITM
[80]	SSBC	-Implements double layered encryption to secure data communications -Efficient Against MITM and Passive Eavesdropping Attacks
[81]	PBBPP	-Analysis of the shortcomings of Pin-Based Authentication Model -Handles Password Guessing Attack
[82]	ESBN	-Developed Link Layer Security Mechanism -Battle against MITM and Message Alterations
[83]	PUF-AP	-Physical Unclonable Function (PUF)-based authentication protocol -Joint authentication and maintenance of secrecy for the session key -MITM, Replay, Passive Eavesdropping, and Impersonation Attacks
[84]	BC-GW	-Blockchain-enabled IoT gateway for BLE-based devices -Preserve user privacy and enforce user preferences when accessing IoT devices

**Table 3 sensors-20-03590-t003:** Summary of Internet of Things (IoT)-Based IDS Systems.

Ref.	Protocol	IDS Method	Methodology	Attacks Mitigated	Results and *Potential Improvements*
[85]	FogComp-IDS	OS-ELM	Used Fog Computing	Cyber	-False Positive Result (FPR) very low, 25% Faster detection rate -*Next step prediction is required to react proactively to the attacks*
[86]	KMA and CBA	KMA with Hash values CBA with path matrix	Technology utilized 6LoWPAN	Routing (Sinkhole and Selective forwarding)	**KMA** 50% to 80% True Positive Result (TPR) **CBA** 76% to 96% TPR -*In depth comparative analysis* -*Only covering few attacks*
[87]	DLM-SLA-IDS	Anomaly-Based	**Dataset** UNSW-NB 15 **Detection Model** -Merged DLM and SLA Algorithm -DLM is Deep auto-encoders -SLA is SVM	General coverage for all attacks	-Proposed method better than other PCA-based and ML methods -*Require more accuracy and low FPR rate*
[88]	LWIDS	Supervised Machine learning-based approach	-Lightweight detection -Used Machine learning based SVM	DoS	Proven that: Good packet arrival rate and SVM based classifier is good for detection *Lacks security parameters policy*
[89]	Three-LIDS	Three layered IDS	-Reporting Normal behavior of nodes -Detect malicious packet -Attack detection	DoS, MITM, reconnaissance and replay	**Accuracy** -Reporting: 96.2% -Malicious Packet: 90% -Attack Detection: 98% *Lacks real time implementation*
[90]	ML-IDS	Machine Learning Based method	Common Protocols Analysis used for SCADA IIOT devices	Backdoor, Command and SQL injection	Capable of handling new attacks like Backdoor, Command, SQL injection *Require hybrid model for better performance*
[92]	MD-CPS	Behavior Rule Specification-based	Unmanned Aerial Vehicle	Zero-day attacks	High detection and prediction *Lacks comparative analysis of other methods and datasets*
[93]	BRIoT	-Behavior Rule Specification-based -For Mission Critical Cyber Physical System -Exploited UAV	-Unmanned Aerial Vehicle -For Mission Critical Cyber Physical Systems	Zero-day attacks	BRIoT outperformed its predecessor BRUIDS *Require more analysis for FPR, FNR viz-a-viz memory, overheads etc*
[94]	InBGG	-InBGG Bayesian-based approach -Gaussian-based	-Feature selection mechanism -Utilized datasets KDDCup’99, KYOTO 2006+, ISCX	Cyber attacks	**InBGG Accuracy** KDDCup’99: 84.06% Kyoto 2006+: 88.13% ISCX: 91.82 **InBGG FPR** KDDCup’99: 16.02% Kyoto 2006+: 13.39% ISCX: 8.37% *Require experimentation with more datasets*
[96]	GLRT	Generalized likelihood ratio test	Three points disturbances detection i.e., unicast packet uplink, downlink and broadcast	Battery Exhaustion and Relay attacks	-Negligible false alarm -Slight missed detection probability *Not good where subgroup of IoT devices are under attack*
[97]	OBSCR	Access Control technique using ontology reasoning	-Analytical vulnerability analysis -Utilized smart meter -Context Inference Rules	Generally, covers all major attacks and Memory dump, Port access Data sniffing, Software Protocol, ZigBee	Proposed system results a. 87.5% b. 91.1% c. 92.5% d. 86.1% e. 78.4% f. 91.5% *More detailed analysis of power system and their vulnerabilities*
[98]	SeArch	Network-based ID (NID)	NID system for SDN-based Cloud IoT	Cyber attacks	Detections: 95.5% Overheads: 8.5% to 15% *Requirement to improve overheads to make IDS power efficient*

**Table 4 sensors-20-03590-t004:** Summary of ZigBee-based IDS.

Ref.	Protocol	Simulation or Testbed	IDS Method	Methodology	Attacks	Results and *Potential Improvements*
[99]	SID-HAN	Simulation (NS2)	Layered Specification-based	IEEE 802.15.4 Standard Normal behaviors	ZigBee Mac and Physical Layer	System capable to detect several known and unknown attacks efficiently *Require:* -*IDS for upper layers of ZigBee* -*Testbed experiments*
[100]	TA-ZB	Simulation (Avrora)	Anomaly-based	-Analysis of KillerBee framework -Implemented IDS in ZigBee radio	KillerBee-Association flooding and packet replay attacks	-Efficient attack detection -*Broad spectrum attack analysis is required*
[101]	ABAS	Testbed	Anomaly-based	Comparison Analysis with Signature and Specification	DoS, Flooding and Pulse DoS	0% FPR for known attacks 95% FPR for unknown attacks
[102]	FL-IDS	Simulation	Fuzzy Logic	Asymptotic Analysis	Flood	-*Require more mathematical and comparison analysis* -*Require analysis of more attacks*
[103]	HAN-IDS-IPS	Simulation	Model-based IDS ML based IPS	Design for PHY and MAC Layers of ZigBee HAN	Cyber Attacks	Efficient in detecting Cyber based attacks *Require Testbed Analysis*
[104]	EE-IDS	Simulation	Optimized watchdog mechanism (OWM)	OWM (trust-based method)	Worm Hole	Good energy consumption, PDR, end-end delay *Proposed system evaluation against mobility models*
[105]	EE-IDSEP	Simulation (NS2)	Energy Efficient Trust	-Developed two modules EE-IDS and EE-IDSEP -Comparison Analysis	Worm Hole and DDOS	Proposed protocol efficiency increased by 23% (AODV), 28% (STR), 33% (OSTR) *Proposed system evaluation against mobility models*
[106]	HHT-TE-IDS	Simulation and Test Bed	HHT and EMD	Scalable HHT-based	LDoS	Capable of detecting Low Rate DoS attack

**Table 5 sensors-20-03590-t005:** Summary of IDS for 6LoWPAN Networks

Ref.	Protocol	Simulation or Testbed	IDS Method	Methodology	Attacks	Results and *Potential Improvements*
[107]	DSD6Lo	Testbed	Signature-based	Improved Open source Suricata IDS	DoS	-Improvement in Suricata IDS in terms of attack detection -IDS run on host computer to make system more resourceful *Require:* -*Real attack implementation* -*Distributed approach and security event databases*
[108]	INTI	Simulation	Watchdog, Reputation and Trust	Combine WR and TB methods	Sinkhole	90% Detection Rate (DR) with Fixed Devices, 70% DR with Mobile Devices
[109]	InDReS	Simulation (NS-2)	Constraint-based specification	Performance Analysis of InDReS with existing system (INTI)	Sinkhole	**Packet Drop Ratio** Lower than INTI **Throughput** Higher than INTI **Packet Delivery Ratio** Higher than INTI **Overhead** Lesser than INTI -*Behavior rule-based analysis* -*Analysis of other attacks*
[110]	CHA-IDS	Simulation (Contiki Cooja) and Testbed	Hybrid (Anomaly +Signature)	-Multi agent system -Feature selection (correlation) -Compared SVELTE, Pongles	Hello Flood Sinkhole Wormhole	99% True Positive Result (TPR)
[111]	ARM	Simulation (Contiki Cooja)	Hybrid Specification -based IDS	Emphasis on RPL	Rank attacks	**Stable Phase** 100% TPR **Instable Phase** 10% FPR 60% Accuracy Rate (AR) -*Require analysis to overcome more attacks*
[112]	ABIDS	Simulation (Contiki Cooja)	Anomaly-based	-Threshold value-based RPL attack detection	DIS and Neighbor	High TPR Low False Positive Result (FPR) *Require more analysis of the RPL attacks* *and other available solutions*
[113]	NSQ	Simulation (Contiki Cooja)	Probabilistic method	Quantitative security assessment of constrained nodes	-Eavesdropping -Data Modification -Blackhole -Selective Forwarding	-High Accuracy -Low power consumption and overheads -Overall efficient system -*Require transmission analysis agent enhancement* -*Real network implementation*

**Table 6 sensors-20-03590-t006:** Summary of IDS for BLE Networks.

Ref.	Protocol	Protocol Specifications	Simulation or Testbed	Methodology	Attacks	Results
[114]	BN-MD	Signature-based IDS	**Testbed** *Attack Node* -Laptop equipped with BackTrack2 *Target Node* -Nokia 6310 Phone (BlueBug Attack) -Sony Ericsson T68i Phone (BlueSnarf Attack) -Plantronics M2500 Headset (CarWhisperer Attack) -Motorola v600 Phone (HeloMoto Attack) *Defense Node* -Hardware Protocol Analyzer -Software IDS Application *Intrusion Response Node* -Three CambridgeSilicon Radio (CSR) chip-based USB Bluetooth dongles.	-IDS using misuse detection -Sytem for detection of Bluetooth Attacks	**Reconnaissance** RFCOMM Scan PSM Scan **DoS** HeaderOverFlow Nasty vCard **Information Theft** BlueSnarf BlueBugger CarWhisperer HeloMoto	**Detection Time (sec)** **Reconnaissance** 110.86 4.74 **DoS** 0.0006 1.10 **Information Theft** 1.46 3.25 0.22 3.22
[115]	BBEADP	IDS and IPS for BLE-based Networks	Test Bed -Broadcom chipset 2070X	Developed an IDS and IPS to increase the overall network life and the throughput	Battery Exhaustion	Approximately 29 h more network life AND 46% more throughput
[116]	BM-IDS	Anomaly-based IDS	Testbed (DK nRF52832, DK nRF52840) -Simulation: BMWatchSim	Method for Optimal Placement of Watchdog for detection	Unknown attacks like Gray hole, Injection or Flooding	**Accuracy** S(best): 100% S(75): 97% S(25): 94%

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
