# Peer review of "Bluetooth Low Energy Mesh Networks: Survey of Communication and Security Protocols"

_sensors, 2020, doi:10.3390/s20123590_

Round 1
Reviewer 1 Report
This is a review paper about three themes concerning BLE mesh networks. The paper appears interesting and comprehensible, and it presents relevant topics.
Unfortunately, there is no mention of how the presented literature was selected; which were the search/inclusion/exclusion criteria for the papers; how did you make sure that your overview of the state of research is complete?
There is a quite high number of acronyms and abbreviations. Please provide a list of acronyms and abbreviations, to increase readability.
In parts, the paper makes the impression of a technical report rather than a journal paper. For instance, a high nesting depth of subsections and subsections of only a few lines should be avoided. There are more suitable presentation forms to express this type of content. I suggest you revise Section 2 and parts of Section 4 with this in mind.
The identification of research gaps yet to be covered seems quite generic. It is unclear how these claims came along. For instance, what is the rationale that more research is needed within "non-flooding based connectionless ... protocols" (line 707)? (this may well be so, but there is no rationale given in the paper). This comment is also valid for most other claims.
Note that the comment of the search criteria is related to the last comment, as you might have missed related research, in case the search criteria did not cover the appropriate topics.
Tables 3 to 6 contain a column "Results and required improvements". The content of the fields of this column is not comprehensible, as they contain internals of the papers referred to in each row. For instance, it is unclear for the reader what "99 percent TPR" means; or "Packet Drop Ration Lower"; or "Little improvement". The way the content of these tables is presented give the impression of a technical report. Please revise this.
Detailed comments:
line 72: incomprehensible: what do you mean by "... may establish a connection ... to initiate ..."
line 74: remove "according to Bluetooth terminology".
line 91: rephrase: This plays a vital role in ...
line 103: rephrase: ... core specifications [8]. In Figure 1 the ...
Figure 3: colour dark green and black cannot be distinguished in print. suggestion to select a different colour.
line 260: something is wrong with this sentence
line 266: is this a repetition? it refers to the same references as in the paragraph above. Please rephrase.
line 295: what do you mean by VHF network? term is not introduced.
line 324: unclear. why are these protocols needed? what do you mean by "overcome these limitations" / "overcome the high overheads" ?
line 329: unclear; also: what is an "orthogonal issue" ?
paragraph starting line 333: vague
line 342: rephrase. "security landscape" is not a good term ...
Section 4.1: these attacks are quite generic. There are no references. Where does this list come from; to what extent is it complete? Are there mitigations to these attacks? To what extent are there already mitigations built-in in BLE?
line 352: which passwords are you referring to? to what extent is this relevant to BLE?
Please also note that this section should be differently presented, avoiding so many subsections.
Figure 6: this figure contains two different things in one: attacks and IDS. note that the term "landscape" is inappropriate in this context. Also the term "issues" is too vague. The caption states that the figure is about threats (not about IDS). I suggest you create two separate drawings. Instead, you could integrate Figure 7 into the appropriate box in Figure 6.
There are some attacks that are not mentioned in your listing. For example, one could think of attacks that destroy the hardware (by overheating) This could, in theory, also hurt people by burning a person's skin, for example in healthcare applications. For the attacks, more literature needs to be referred to; there is a vast corpus, including review-articles to such attacks. Please refer to these.
line 462: "It resembles" ... vague. To what extent is it similar?
The list of security enhancements in Section 4.4 is not systematic. There might be more work on such enhancements in the literature
Line 502: in contrast to what?
line 510: Table 2 shows a summary ... ; what do you mean by efforts? please rephrase.
Table 2, second entry: what do you mean by "good for" DoS and Relay attacks? does it support attacks? I doubt that you want to express this. please rephrase.
line 518: remove "Landscape of". The term landscape is not suitable in this context.
The section about IDS contains references to technologies that are more generic than or different to BLE mesh networks. Please give a rationale why this is relevant in your paper before Section 5.1.
Tables 3 and following are difficult to understand, as already mentioned above.
Section 5.2: please give a rationale in the paper what the relation between Zigbee and BLE is.
line 641: suggestion: please rephrase. using capitalising to show the acronym looks strange in this context.
line 717: remove "Nonetheless, "
Why are some BLE mesh network concepts and security features in appendices? Please consider integrating these into the main text. When integrating, please avoid many short sections in the presentation.
Author Response
To: The Editor
Reply to reviewers on sensors-809625
We have attached our updated manuscript, and our point-by-point response to the comments as given below.
Reviewer 1
- This is a review paper about three themes concerning BLE mesh networks. The paper appears interesting and comprehensible, and it presents relevant topics."
Reply: We are very grateful for your kind comments.
- Unfortunately, there is no mention of how the presented literature was selected; which were the search/inclusion/exclusion criteria for the papers; how did you make sure that your overview of the state of research is complete?
Reply: We agree.
Action: We have added an explanation in the “Introduction” section (New Line Number: 27-30).
- In parts, the paper makes the impression of a technical report rather than a journal paper. For instance, a high nesting depth of subsections and subsections of only a few lines should be avoided. There are more suitable presentation forms to express this type of content. I suggest you revise Section 2 and parts of Section 4 with this in mind.
Reply: We agree.
Action: We have improved the presentation of Section 2 and 4.
- In The identification of research gaps yet to be covered seems quite generic. It is unclear how these claims came along. For instance, what is the rationale that more research is needed within "non-flooding based connectionless ... protocols" (line 707)? (this may well be so, but there is no rationale given in the paper). This comment is also valid for most other claims.
Reply: We have identified the extracted research gaps in Section 3.4 and 5.5 after the detailed analysis of BLE Mesh Protocols covered in first part of the paper.
- Tables 3 to 6 contain a column "Results and required improvements". The content of the fields of this column is not comprehensible, as they contain internals of the papers referred to in each row. For instance, it is unclear for the reader what "99 percent TPR" means; or "Packet Drop Ration Lower"; or "Little improvement". The way the content of these tables is presented give the impression of a technical report. Please revise this.
Reply: We agree.
Action: We have revised the tables 3 to 6. Also, we have put the explanations where possible.
- line 72: incomprehensible: what do you mean by "... may establish a connection ... to initiate ..."
Reply: We agree.
Action: We have rephrased the sentence (New Line Number: 85-89).
- line 74: remove "according to Bluetooth terminology".
Reply: We agree.
Action: We have removed it (New Line Number: 88)
- line 91: rephrase: This plays a vital role in ....
Reply: We agree.
Action: We have rephrased the sentence (New Line Number: 67).
- line 103: rephrase: ... core specifications [8]. In Figure 1 the ...
Reply: With due respect, we have left as it is as we are referring to the Figure 1 that is depicting the BLE Mesh System Architecture defined on top of Bluetooth Core Specifications.
- Figure 3: colour dark green and black cannot be distinguished in print. suggestion to select a different colour.
Reply: We agree.
Action: We have changed the colors of the Figure 3 (Page Number 7).
- line 260: something is wrong with this sentence.
Reply: We agree.
Action: We have rephrased it (New Line Number: 262).
- line 266: is this a repetition? it refers to the same references as in the paragraph above. Please rephrase.
Reply: With due respect, there is no repetition. The author of both the papers is same and we have referenced his latest paper before his old paper. And on the next paragraph we discuss the author’s latest paper (New Line Number: 269).
- line 295: what do you mean by VHF network? term is not introduced.
Reply: We agree.
Action: We have explained the term (New Line Number: 297).
- line 324: unclear. why are these protocols needed? what do you mean by "overcome these limitations" / "overcome the high overheads" ?
Reply: We agree.
Action: We have rephrased it (New Line Number: 328-332).
- line 329: unclear; also: what is an "orthogonal issue" ?
Reply: We agree.
Action: We have rephrased it for clarity (333-337).
- paragraph starting line 333: vague?
Reply: We agree.
Action: We have rephrased (New Line Number: 338-339).
- line 342: rephrase. "security landscape" is not a good term ...?
Reply: We agree.
Action: We have rephrased it (New Line Number: 347).
- Section 4.1: these attacks are quite generic. There are no references. Where does this list come from; to what extent is it complete? Are there mitigations to these attacks? To what extent are there already mitigations built-in in BLE?
Reply: We agree.
Action: We have put the references. Moreover, we have provided the available BLE security features in Appendix section where we have discussed the mitigations as well.
- line 352: which passwords are you referring to? to what extent is this relevant to BLE?
Reply: We agree.
Action: We are referring to the PIN/Password utilized by BLE for authentication.
- Please also note that this section should be differently presented, avoiding so many subsections.?
Reply: We agree.
Action: We have improved the presentation of the section (Page 11-14).
- Figure 6: this figure contains two different things in one: attacks and IDS. note that the term "landscape" is inappropriate in this context. Also, the term "issues" is too vague. The caption states that the figure is about threats (not about IDS). I suggest you create two separate drawings. Instead, you could integrate Figure 7 into the appropriate box in Figure 6.?
Reply: We agree.
Action: We have combined the Figure 6 and Figure 7 of previous version of the paper and eliminated IDS part from Figure 6 and shown in a separate Figure (in Fig. 8 of new version of the paper).
- There are some attacks that are not mentioned in your listing. For example, one could think of attacks that destroy the hardware (by overheating) This could, in theory, also hurt people by burning a person's skin, for example in healthcare applications. For the attacks, more literature needs to be referred to; there is a vast corpus, including review-articles to such attacks. Please refer to these.
Reply: The paper focused on network-based attacks.
- line 462: "It resembles" ... vague. To what extent is it similar?
Reply: We agree.
Action: We have rephrased (New Line Number 467).
- The list of security enhancements in Section 4.4 is not systematic. There might be more work on such enhancements in the literature?
Reply: We agree. We have emphasized the works which are relevant to our paper. Also, BLE 5 has adopted improved security features.
- Line 502: in contrast to what?
Reply: We agree.
Action: We have rephrased it (New Line Number: 508)
- line 510: Table 2 shows a summary ...; what do you mean by efforts? please rephrase.
Reply: We agree.
Action: We have rephrased it (New Line Number: 516)
- Table 2, second entry: what do you mean by "good for" DoS and Relay attacks? does it support attacks? I doubt that you want to express this. please rephrase.
Reply: We agree.
Action: We have rephrased it.
- line 518: remove "Landscape of". The term landscape is not suitable in this context.
Reply: We agree.
Action: We have rephrased (New Line Number: 524).
- The section about IDS contains references to technologies that are more generic than or different to BLE mesh networks. Please give a rationale why this is relevant in your paper before Section 5.1.
Reply: We have provided a justification in New Line Number: 525-529.
- Tables 3 and following are difficult to understand, as already mentioned above.
Reply: We agree.
Action: We have improved accordingly.
- Section 5.2: please give a rationale in the paper what the relation between Zigbee and BLE is.
Reply: We agree.
Action: We have improved accordingly as ZigBee is a competing technology to BLE in terms of coverage and use cases (New Line Number: 605-606).
- line 641: suggestion: please rephrase. using capitalising to show the acronym looks strange in this context.
Reply: We agree.
Action: We have rephrased it (New Line Number: 647).
- line 717: remove "Nonetheless, ".
Reply: We agree.
Action: We have removed it (New Line Number: 729).
- Why are some BLE mesh network concepts and security features in appendices? Please consider integrating these into the main text. When integrating, please avoid many short sections in the presentation.
Reply: We have kept general information on BLE in the Appendix section to reduce the main paper length as it is considered background information on BLE technology. Nonetheless, it is provided as a reference for readers.

Reviewer 2 Report
Good extensive survey with plenty of references.
Please read the paper for improved readability. There are few places that can be improved, for in example line 4, increase should be changed to increased.
Author Response
To: The Editor
Reply to reviewers on sensors-809625
We have attached our updated manuscript, and our point-by-point response to the comments as given below.
Reviewer 2
- Good extensive survey with plenty of references.
Reply: We are very grateful for your kind comments.
- Please read the paper for improved readability. There are few places that can be improved, for in example line 4, increase should be changed to increased.
Reply: We agree.
Action: We have improved accordingly.

Reviewer 3 Report
This paper reviews the most recent BLE-based mesh network protocols and related security issues.
The subject of the paper is worthy of investigation, and the paper is well written. I quite liked reading it. However, it requires some revisions and additional experiments before being accepted for publication. Here are my comments:
Major concern:
My biggest concern is power consumption and delay in various Mesh protocols. Energy modeling is a crucial element in wireless network simulation, and energy consumption is an essential metric for evaluating the performance of wireless network protocols. The authors provide some data in Table 1, but not for all examined protocols. I presume that authors have just reported the delay and energy from the original papers. But in my opinion, this is not sufficient in a survey paper. The authors should provide the estimates for delay and energy consumption for all examined protocols. This can be done by actually measuring the energy consumption or by selecting an appropriate energy model and running a simulation.
I suggest the authors look into some recently published papers on BLE energy consumptions to get an idea of how to measure or estimate energy consumption and delay, e.g.,
- Nikodem, M.; Bawiec, M. Experimental Evaluation of Advertisement-Based Bluetooth Low Energy Communication. Sensors 2019, 20, 107.
- Bulić, P.; Kojek, G.; Biasizzo, A. Data Transmission Efficiency in Bluetooth Low Energy Versions. Sensors 2019, 19, 3746.
- Basu, S.S.; Haxhibeqiri, J.; Baert, M.; Moons, B.; Karaagac, A.; Crombez, P.; Camerlynck, P.; Hoebeke, J. An End-To-End LwM2M-Based Communication Architecture for Multimodal NB-IoT/BLE Devices. Sensors 2020, 20, 2239.
Minor comment:
Tables are hardly readable due to small fonts. The authors should put more effort into improving the readability of the tables in the paper.
Author Response
To: The Editor
Reply to reviewers on sensors-809625
We have attached our updated manuscript, and our point-by-point response to the comments as given below.
Reviewer 3
- The subject of the paper is worthy of investigation, and the paper is well written. I quite liked reading it. However, it requires some revisions and additional experiments before being accepted for publication.
Reply: We are very grateful for your kind comments.
- My biggest concern is power consumption and delay in various Mesh protocols. Energy modeling is a crucial element in wireless network simulation, and energy consumption is an essential metric for evaluating the performance of wireless network protocols. The authors provide some data in Table 1, but not for all examined protocols. I presume that authors have just reported the delay and energy from the original papers. But in my opinion, this is not sufficient in a survey paper. The authors should provide the estimates for delay and energy consumption for all examined protocols. This can be done by actually measuring the energy consumption or by selecting an appropriate energy model and running a simulation
Reply: With due respect, this paper is a literature survey of existing works where reported results are used as the basis for evaluation, as documented in Table 1. Simulation and experimental evaluation of the surveyed work is out of scope for the paper and should be addressed separately.
Nonetheless, we have indicated the research gap in terms of the need to evaluate energy efficiency performance of BLE Mesh Protocols in the "Conclusion" section of the paper (New Line Number: 716 to 721).
- Tables are hardly readable due to small fonts. The authors should put more effort into improving the readability of the tables in the paper.
Reply: We agree.
Action: We have improved them.

Reviewer 4 Report
The article is interesting but too long and with plenty of more than 100 abbreviations. It is difficult to interpret and follow the line of thought of the article. There are abbreviations that only occur once, so they are completely unnecessary.
Figure 5 is your own figure or is it adapted from somewhere?
The letters of the tables are too small. It is very hard to read them.
Page 13, line 418: What is CPAL abbreviation?
Page 13, line 419: What is GLARM abbreviation?
14th page, line 432: LTK abbreviation is used but it is defined only on the 23rd page, line 775.
15th page line 482: ECDH abbreviation is used but it is defined only on the 24th page, lines 788-789.
17th page, line 547: „SVM” is written. Is this abbreviation support-vector machines?
To sum it up the paper is too long, it could be divided into 2 papers, moreover, the number of abbreviations is very huge. See the list below:
Bluetooth Low Energy (BLE)
IntrusionDetectionSystems(IDS)
Wireless Ad-Hoc Network (WAHN)
Personal Area Network (PAN)
Intrusion Detection Systems (IDS)
Generic Access Profile (GAP)
Generic Attribute Profile (GATT)
Security Manager (SM)
Attribute Protocol (ATT)
Logical Link Control and Adaptation62 Protocol (L2CAP)
Link Layer (LL)
HostControlInterface(HCI)
Adaptive Frequency Hopping (AFH)
On-Demand Distance Vector (AODV)
Dynamic Source Control Routing (DSR)
Optimized Link State Routing (OLSR)
Destination Sequenced Distance Vector (DSDV)
Distance Routing Effect Algorithm for Mobility (DREAM)
Better Approach To Mobile Adhoc Networking (BATMAN)
Two-Tier Data Dissemination Protocol (TTDD)
Energy Efficient Secured Ring Routing (E2SR2)
Intelligent and Secured Fuzzy Clustering Algorithm Using Balanced Load Sub-Cluster Formation (ISFC-BLS)
Scalable Energy Efficient Clustering Hierarchy protocol (SEECH)
Multi-objective Fuzzy Clustering Algorithm (MOFCA)
MultiHop Transfer Service (MHTS)
Packet Delivery Ratio (PDR)
Named Data Networking (NDN)
Software Defined Network (SDN)
best-effort scheduling (BES)
bounded flooding (RBF)
data-channel-based BLE mesh networks (DC-BMN)
Directional Ad-Hoc On-demand Multipath Distance Vector (D-AOMDV)
heterogeneous intelligent robotic network (HIRO-NET)
Power Line Communications (PLC)
Wireless Personal Area Network (WPAN)
Man in the Middle (MITM)
Media Access Control (MAC)
Network Interface Card (NIC)
Denial of Service (DoS)
Long Term Key (LTK)
Key Negotiation of Bluetooth (KNOB)
Bluetooth Topology Construction Protocol (BTCP)
Elliptic-curve Diffie–Hellman (ECDH)
Secure and Simple Pairing (SSP)
Hash-based Message Authentication Code (HMAC)
Numeric Comparison (NC)
Physical Unclonable Function (PUF)
Instursion Detection Systems (IDS)
network operation centre (NOC)
OnlineSequential-Extreme LearningMachine(OS-ELM)
Key-Match (KMA)
Cluster-Based (CBA)
deep-learning method (DLM)
shallow-learning approach (SLA)
Support Vector Machine (SVM)
Artificial Bee Colony (ABC)
Five-foldCrossValidation(5FCV)
Principal Component Analysis (PCA)
Machine Learning (ML)
Neural Network (NN)
k-Nearest Neighbor (k-NN)
Device Tree (DT)
Supervisory Control And Data Acquisition (SCADA)
Industrial Internet of Things (IIoT)
Rule-basedintrusiondetectionmethodology(BRIoT)
Infinite Bounded Generalized Gaussian mixture (InBGG)
False Positive Rate (FPR)
GeneralizedLikelihoodRatioTest (GLRT)
software-defined networking (SDN)
network function virtualization (NFV)
Home Area Networks (HAN)
Energy-Efficient IDS with Energy Prediction (EE-IDSEP)
Ad-hoc On-Demand Distance Vector (AODV)
Shortcut Tree Routing (STR)
Opportunistic Shortcut Tree Routing (OSTR)
Energy Efficient Trust System (EE-TS)
Low-rate DoS (LDoS)
Hilbert-Huang Transforms (HHT)
Trust Evaluation (TE)
Internet Engineering Task Force (IETF)
Detection and Response System (InDRES)
Compression Header Analyzer Intrusion Detection System (CHA-IDS)
Authenticated Rank and routing Metric (ARM)
RPL(RoutingProtocolforLow-power andlossynetworks)
IPv6overTime-SlottedChannelHopping(6TiSCH)
Node Security Quantification (NSQ)
Message Security Value (MSV)
Damage Level (DL)
Low Energy Legacy Pairing (LLP)
Long Term Key (LTK)
Transport Protocol (TP)
Identity Resolving Key (IRK)
Connection Signature Resolving Key (CSRK)
Out of Band(OOB)
TemporaryKey(TK)
Low Energy Secure Connection (LESC
Short Term Key (STK)
Elliptic-curve Diffie–Hellman (ECDH)
Near Field Communications (NFC)
Pass Key Entry (PKE)
Just Work (JW)
Man in the Middle (MITM)
Numeric Comparison (NC)
Author Response
To: The Editor
Reply to reviewers on sensors-809625
We have attached our updated manuscript, and our point-by-point response to the comments as given below.
Reviewer 4
- The article is interesting but too long and with plenty of more than 100 abbreviations. It is difficult to interpret and follow the line of thought of the article. There are abbreviations that only occur once, so they are completely unnecessary.
Reply: We are very grateful for your kind comments. Moreover, we have improved the paper readability in terms of abbreviations.
- Figure 5 is your own figure or is it adapted from somewhere?
Reply: We have referenced the figure.
- The letters of the tables are too small. It is very hard to read them.
Reply: We agree.
Action: We have improved them where possible.

Round 2
Reviewer 1 Report
The authors have addressed most of the comments sufficiently. I have some minor comments:
Section 2 is still written with many numbered paragraphs. However, in principle, the authors present a list of architecture elements. Thus, a different way of presenting the content would be suitable.
line 80: incomprehensible: "see advertisements". Should that be "sees"?
In Table 1, you use C-O, R, and F. Would it be better to mark the connection-oriented entries with C instead of C-O? Further, you introduce some further values (such as TR) in the table, instead of the header. Note also that FM in Entry 2 is first introduced in Entry 5.
Typo in the box of Figure 7. In the box on top, the word "Sweyn" is misspelt.
Tables 3 and following. In the right column, the headline: what do you mean by "required improvements"? Do you mean "observed" improvements? Or "claimed" improvements? Or "supposed", "putative", "expected" ... improvements?
Line 710: better to use "survey" instead of "paper".
I suggest you add a sentence at the end, after line 738, where you conclude in which areas which research questions still are open.
I still think that it would be better to integrate the content of the appendices into the main text.
The paper contains has a large number of abbreviations and acronyms. Please add a list of these, as also suggested in the template. Such a list would increase the readability of the paper, and would also be a contribution in itself, collecting the used abbreviations in one table (as a kind of dictionary).
Author Response
To: The Editor
Reply to the Reviewer 1 after Revised Version Review of sensors-809625
We would like to thank the editor (including concerned staff) and the respectable Reviewer for giving very quick response and valuable guidance. The paper has now again been corrected as per the comments of Reviewer 1 given after revised version review.
Reviewer 1
(Minor Comments)
- The authors have addressed most of the comments sufficiently.
Reply: We are very grateful for your kind comment.
- Section 2 is still written with many numbered paragraphs. However, in principle, the authors present a list of architecture elements. Thus, a different way of presenting the content would be suitable.
Reply: We agree.
Action: We have further improved the presentation of the Section.
- line 80: incomprehensible: "see advertisements". Should that be "sees"?
Reply: We agree.
Action: We have corrected it.
- In Table 1, you use C-O, R, and F. Would it be better to mark the connection-oriented entries with C instead of C-O? Further, you introduce some further values (such as TR) in the table, instead of the header. Note also that FM in Entry 2 is first introduced in Entry 5.
Reply: We agree.
Action: We have corrected it.
- Typo in the box of Figure 7. In the box on top, the word "Sweyn" is misspelt.
Reply: We agree.
Action: We have corrected it.
- Tables 3 and following. In the right column, the headline: what do you mean by "required improvements"? Do you mean "observed" improvements? Or "claimed" improvements? Or "supposed", "putative", "expected" ... improvements?"
Reply: We agree.
Action: We have changed to “Potential Improvements”.
- Line 710: better to use "survey" instead of "paper".
Reply: We agree.
Action: We have corrected it (New Line Number: 698 and 710).
- I suggest you add a sentence at the end, after line 738, where you conclude in which areas which research questions still are open.
Reply: We agree.
Action: We have added a sentence (New Line Number: 727-730).
“In short, energy efficient BLE pure mesh protocol capable of multicasting and topology auto-configuration is needed to support scalable applications effectively. In addition, intelligent IDS that can protect the BLE mesh networks from attacks are also crucial for the secure operation of these applications.”
- I still think that it would be better to integrate the content of the appendices into the main text.
Reply: With due respect, the information in the Appendix is background information that is not directly needed to understand the contents of the survey, and is provided as additional information for readers who are not so familiar with the topic.
- The paper contains has a large number of abbreviations and acronyms. Please add a list of these, as also suggested in the template. Such a list would increase the readability of the paper, and would also be a contribution in itself, collecting the used abbreviations in one table (as a kind of dictionary).
Reply: We agree.
Action: We have added the List of Abbreviations.

Reviewer 4 Report
Thank you for correcting the manuscript based on the reviewers' comments. In spite of the fact that this version is better than the earlier one, the number of abbreviations is still huge.
Author Response
To: The Editor
Reply to the Reviewer 4 after Revised Version Review of sensors-809625
We would like to thank the editor (including concerned staff) and the respectable Reviewer for giving very quick response and valuable guidance. The paper has now again been corrected as per the comments of Reviewer 4 given after revised version review.
Reviewer 4
(Minor Comments)
- Thank you for correcting the manuscript based on the reviewers' comments. In spite of the fact that this version is better than the earlier one, the number of abbreviations is still huge.
Reply: We are very grateful for your kind comments. We have improved the paper readability by adding List of Abbreviations.
